# Modulation of JAK-STAT Signaling by LNK: A Forgotten Oncogenic Pathway in Hormone Receptor-Positive Breast Cancer

**DOI:** 10.3390/ijms241914777

**Published:** 2023-09-30

**Authors:** José A. López-Mejía, Jessica C. Mantilla-Ollarves, Leticia Rocha-Zavaleta

**Affiliations:** 1Departamento de Biología Molecular y Biotecnología, Instituto de Investigaciones Biomédicas, Universidad Nacional Autónoma de México, Mexico City 03100, Mexico; andres.lopezmj@gmail.com (J.A.L.-M.); jessimantilla525@gmail.com (J.C.M.-O.); 2Programa Institucional de Cáncer de Mama, Instituto de Investigaciones Biomédicas, Universidad Nacional Autónoma de México, Mexico City 03100, Mexico

**Keywords:** hormone receptor-positive breast cancer, JAK-STAT, LNK, IL-6, prolactin, oncogenic pathways

## Abstract

Breast cancer remains the most frequently diagnosed cancer in women worldwide. Tumors that express hormone receptors account for 75% of all cases. Understanding alternative signaling cascades is important for finding new therapeutic targets for hormone receptor-positive breast cancer patients. JAK-STAT signaling is commonly activated in hormone receptor-positive breast tumors, inducing inflammation, proliferation, migration, and treatment resistance in cancer cells. In hormone receptor-positive breast cancer, the JAK-STAT cascade is stimulated by hormones and cytokines, such as prolactin and IL-6. In normal cells, JAK-STAT is inhibited by the action of the adaptor protein, LNK. However, the role of LNK in breast tumors is not fully understood. This review compiles published reports on the expression and activation of the JAK-STAT pathway by IL-6 and prolactin and potential inhibition of the cascade by LNK in hormone receptor-positive breast cancer. Additionally, it includes analyses of available datasets to determine the level of expression of LNK and various members of the JAK-STAT family for the purpose of establishing associations between expression and clinical outcomes. Together, experimental evidence and in silico studies provide a better understanding of the potential implications of the JAK-STAT-LNK loop in hormone receptor-positive breast cancer progression.

## 1. Introduction

Breast cancer is by far the most commonly diagnosed neoplasia in women worldwide, accounting for more than 25% of all female cancer cases reported in 2020. The number of estimated new breast cancer cases is 2.6-times higher than that estimated for colorectal cancer, which is the second most common cancer in women [1].

The mammary gland is a hormone-dependent organ. Mammary cell development, differentiation, and function depend basically on the action of ovarian steroid hormones estrogen and progesterone, along with prolactin. Estrogen-activated signal transduction has been demonstrated to be essential for the development of the gland [2], while progesterone and prolactin induce the differentiation of milk-producing cells during late pregnancy [3]. The activity of estrogen and progesterone is mediated by binding to their specific receptors. Estrogen receptor (ER) and progesterone receptor (PR) are members of the nuclear receptor superfamily [4]. They are transcription factors that induce the expression of several genes to regulate not only the function of ER+/PR+ cells but also that of ER-/PR- cells, establishing paracrine networks to support normal mammary gland development [3]. Unsurprisingly, the deregulation of ER/PR activity is a driver of carcinogenesis in mammary gland cells.

ER and PR are considered prognostic markers of and predictive factors for breast tumor hormonal therapy. As such, ER/PR expression is routinely assessed in order to establish the treatment that will provide the greatest benefit to the patient. Endocrine therapy has focused on blocking ER activity or inhibiting the aromatase-mediated conversion of androgens into estrogens [5]. Unfortunately, endocrine therapy resistance has developed in a proportion of patients. Intrinsic resistance to some hormonal therapy has been documented in early-stage and metastatic disease patients. Patients with metastasis may also present recurrence soon after beginning adjuvant hormonal treatment. Moreover, resistance can also be acquired after successful initial treatment, followed by short responses to serial hormonal therapies until the tumor becomes refractory [6]. Even the application of combined therapeutic approaches, using aromatase inhibitors along with CDK4/6-interfering small molecules or inhibitors of the PI3K/AKT/mTOR pathway, has shown limited benefit because of the intrinsic or eventual development of resistance in a proportion of patients [6]. For these reasons, there is a need to identify new targets in resistant tumors in order to provide patients with new therapeutic options. The study of alternative cancer-associated signaling cascades, such as the Janus kinase (JAK) signal transducer and activator of transcription (STAT) pathway, may provide new targets for breast cancer treatment.

The JAK-STAT cascade has been found to be constitutively active in breast cancer and has been proposed as a modulator of chemotherapy resistance [7]. Moreover, inhibitors of various elements of the pathway are undergoing testing, and they appear to be promising drugs for the treatment of breast cancer [8]. In healthy cells, JAK-STAT signaling is regulated by the lymphocyte adaptor protein (LNK). LNK belongs to the Src homology 2 (SH2) domain-containing adaptor protein family. This controls signal transduction pathways downstream of various growth factors, cytokine receptors, and receptor tyrosine kinases (RTKs) by inhibiting JAK kinase activity [9]. For this reason, LNK has been investigated in cancer cells as a potential regulator of JAK-STAT carcinogenic activities [9].

In this review, we aim to overview JAK-STAT signaling in hormone receptor-positive breast cancer as well as the participation of LNK in the regulation of this signal transduction pathway. In addition, we aim to integrate available datasets in order to better understand the potential association of JAK-STAT and LNK in the course of hormone receptor-positive breast cancer.

## 2. Hormone Receptor-Positive Breast Cancer

Studies estimating future breast cancer incidence predict an increasing global burden. A clear pattern of increasing breast cancer mortality rates has been detected in many low- and middle-income countries with fragile health systems. These health systems can lack the comprehensive national strategies required for early diagnosis, access to proper treatments, and palliative care [10,11,12]. In addition, it is expected that inequities in breast cancer mortality will be intensified as a result of the COVID-19 pandemic. A population-based modeling study predicted a 7.9–9.6% increase in breast cancer deaths up to year 5 after diagnosis compared with pre-pandemic figures [13]. In order improve the clinical management of breast cancer, tumors are classified by histologic type [14,15] and by the expression of human epidermal growth factor receptor 2 (HER2) [16,17], PR, and ER [18] into four intrinsic subtypes: luminal A, luminal B, HER2-enriched, and basal-like breast cancer [19,20,21]. The basic immunohistochemical [22] and genetic [23] characteristics along with the frequency and treatment [24,25,26] of breast cancer subtypes are shown in Table 1.

Since estrogens are the leading molecules inducing the deregulation of cell activity by binding to ERs in luminal tumors, undergoing hormone therapy is considered a risk factor for this type of cancer [27,28]. To date, three main receptors have been identified: ERα, ERβ, and the non-classical G-protein-coupled estrogen receptor 1 (GPER1). ERα and ERβ are intracellular receptors that bind estrogens in the cytoplasmic compartment, forming a complex that eventually translocates into the nucleus. The complex then recognizes and binds to estrogen response elements (EREs) in target gene promoters, inducing gene transcription. In contrast, GPER1 is a transmembrane protein that interacts with membrane-anchored estrogens, prompting the activation of various cell signaling cascades, along with the production of cyclic adenosine monophosphate (cAMP) [29].

The first ER identified was ERα [30]. ERα is significantly overexpressed in breast cancer as compared with normal breast tissue [31]. ERα’s carcinogenic effect is the result of the unique array of stimulated genes that modulate cell proliferation and differentiation, in addition to providing protection against cell apoptosis [32,33,34,35]. Although ERβ was initially associated with breast cancer progression [36,37], a further report studying the intracellular localization of ERβ variants via immunofluorescence in tumor samples provided evidence showing that nuclear expression of ERβ2 and ERβ5 variants correlated with increased overall and disease-free survival [38]. ERβ2 and ERβ5 can jointly form stable heterodimers with ERα [39], suggesting that heterodimerization may inhibit ERα-mediated gene expression, thus resulting in favorable prognosis [40]. On the other hand, GPER1 is a G-protein-coupled membrane receptor [41] with weak affinity for binding estrogens [42]. It is expressed in ER-positive and ER-negative breast tumors [43,44]. After binding to GPER1, estrogens stimulate mitogen-activated protein kinase (MAPK) [45] and steroid receptor coactivator (SRC) tyrosine kinase, establishing complex crosstalk with epidermal growth factor receptor (EGFR) [46,47]. The clinical relevance of GPER1 expression remains controversial, with some data showing that the downregulation of GPER1 is associated with cancer progression [48], while others suggest that lack of GPER1 expression is a good prognosis factor [49]. Moreover, GPER1 expression has been associated with resistance to tamoxifen treatment [50], but some reports indicate that GPER1 is able to hamper migration and angiogenesis [51,52]. Clearly, the importance of GPER1 in breast cancer is still to be ascertained.

## 3. JAK-STAT Signaling Pathway

JAK-STAT is an essential pathway that transduces signals downstream of more than 50 growth factors, hormones, interferons, and interleukins [53]. The JAK-STAT signal is activated after ligands are bound to classical receptors and RTKs. JAKs are noncovalently bound to receptor intracellular domains. After ligand binding, the kinase activity of JAKs is activated and they phosphorylate tyrosine residues of cytoplasmic receptor regions. Phosphorylated receptor residues provide docking sites for STAT recruitment via their SH2 domains. When STATs bind to the receptor, they are phosphorylated onto tyrosine residues, allowing the formation of homo- and heterodimers. STAT dimers translocate into the nucleus to undergo binding and activate the transcription of target genes [54]. The cellular events regulated by JAK-STAT are diverse and comprise inflammation, apoptosis, tissue repair, immune responses, hematopoiesis, and adipogenesis [55]. Thus, failure to control the JAK/STAT cascade leads to diverse human illnesses.

### 3.1. The JAK Family

The JAK family consists of four members: JAK1, JAK2, JAK3, and TYK2. JAKs are tyrosine kinases that share a conserved carboxy terminus kinase domain, a pseudokinase domain that regulates kinase activity, SH2, and a FERM domain that participate together in the union of JAKs with cell receptors [56,57,58,59].

JAK1 is expressed extensively in mammalian cells, and its activation is mediated by many receptors on immune system cells. Accordingly, IL-2, IL-3, IL-4, IL-5, IL-6, IL-7, IL-9, IL-10, IL-11, IL-15, IFNα, IFNβ, IFNγ, and leukemia inhibitor factor (LIF) receptors initiate JAK1-induced activation of a signaling cascade that regulates immune responses [60].

Like JAK1, JAK2 can activate signals downstream of IFNα, IFNβ, IFNγ, LIF, IL-3, IL-5, IL-6, IL-11, and granulocyte-macrophage colony-stimulating factor (GM-CSF) receptors. In addition, JAK2 has been found to be associated with erythropoietin (EPO) receptor, thrombopoietin receptor, growth hormone receptor, and prolactin (PRL) receptor [61], indicating that JAK2 is implicated in a wide range of cell functions.

On the other hand, JAK3 participates in the activation of signals that are mediated by cytokine receptors with the γc receptor subunit, such as IL-2, IL-4, IL-7, IL-9, IL-15, and IL-21 receptors [62]. Finally, TYK2 is implicated in regulating Th1/Th2 balance during allergic reactions [63] by activating signals from IL-6, IL-10, IL-12, IL-13, and IL-23 receptors [64].

### 3.2. The STAT Family

The family of STAT transcription factors is composed of seven different proteins: STAT1, STAT2, STAT3, STAT4, STAT5A, STAT5B, and STAT6. Structurally, they share an N-terminal domain, a coiled-coil domain, a DNA-binding domain, a conserved SH2 domain, and a transcription activation domain. Inactive STATs are found in the cytoplasmic compartment as anti-parallel dimers; upon activation by JAK-mediated phosphorylation, STAT dimers form parallel, nutcracker-like dimeric structures. Active STAT dimers translocate into the nucleus, where they recognize and bind to the palindromic DNA sequence TTCN_2-4_GAA in promoters and enhancers of target genes, thus inducing transcription [65].

STAT1 is activated as a response to cytokines like IFNs, IL-2, IL-6, platelet-derived growth factor (PDGF), epidermal growth factor (EGF), hepatocyte growth factor, tumor necrosis factor (TNF), and angiotensin II. STAT1 is able to promote the expression of genes that inhibit the cell cycle [66], regulate cell differentiation [67], induce apoptosis [68,69], and regulate important activities of the immune system [70]. The function landscape associated with STAT1 activation categorizes this transcription factor as a potential tumor inhibitor.

STAT2 is basically involved in regulating immune responses induced by IFNα and IFNβ, including anti-viral reactions [71] and the generation of memory cells [72].

STAT3 activation is induced by several immune-related factors, including members of the IL-6 family (IL-6, IL-11, IL-31, LIF, etc.), members of the IL-10 family (IL-10, IL-19, IL-20, IL-22, IL-24, and IL-26), granulocyte colony-stimulating factor (G-CSF), and IFNs [73]. In conjunction with its participation as an immune and inflammatory response regulator, STAT3 has been reported to be a pro-tumoral transcription factor. Experimental evidence supporting the participation of STAT3 in cancer progression will be further described.

STAT4 transcription activity is stimulated by IFNα, IFNβ, IL-12, and IL-23 [74]. STAT4 is fundamental for the differentiation of Th1 cells, the maturation of B lymphocytes, and for promoting immunoglobulin switch during humoral responses [75].

STAT5A and STAT5B share 91% of amino acid residues. They are activated by immune-associated cytokines such as IL-2, IL-4, IL-7, IL-9, and IL-15. In addition, STAT5 responds to the stimulation of growth factors like EGF, PDGF, GM-CSF, and erythropoietin (EPO) [76]. The variety of stimuli activating STAT5 contributes to its wide range of biological functions. Accordingly, STAT5 has been reported to participate in the proliferation and activation of T lymphocytes and natural killer cells (NK) [77]. STAT5 is also involved in cell proliferation and apoptosis [78]. In addition, STAT5 is an important regulator of mammary gland development [79]. This is the reason why it also participates in the development and progression of breast cancer. The role of STAT5 in breast cancer will be further described.

STAT6 transduces signals from the receptors of IL-4 and IL-13 [80]. Therefore, the transcriptional activity of STAT6 controls Th2 lymphocyte differentiation [81]. Furthermore, STAT6 regulates B cell proliferation, immunoglobulin isotype switching, and production of IgE [82].

The JAK-STAT transduction pathway is crucial for the development and correct functioning of a number of cells; thus, the loss of JAK/STAT regulation is expected to be involved in various human disorders. In fact, there is reliable evidence supporting the participation of JAK-STAT in different types of tumors, including breast cancer.

## 4. JAK-STAT Signaling in Hormone Receptor-Positive Breast Cancer

Hormone receptor-positive breast cancer has been traditionally associated with ERα-mediated activation of the PI3K-AKT and Ras-MAPK cascades. However, there is increasing evidence showing that other signaling pathways can also be activated and participate in providing growing advantages to tumor cells. A relevant alternative signaling cascade in hormone receptor-positive breast cancer is the JAK-STAT pathway, which is associated with increasing cell proliferation and the acquisition of resistance to treatment. STAT3 has been found to be constitutively activated in a high proportion of all breast cancer subtypes [83]. As detailed above, JAK-STAT signaling can be activated by a series of receptors. In the case of luminal breast cancer, two meaningful activators have been described: IL-6 receptor (IL-6R) and PRL receptor (PRLR).

### 4.1. Activation of the JAK-STAT Pathway by IL-6

IL-6 is a cytokine that mediates inflammatory responses by activating JAK2/STAT3 signaling. Likewise, IL-6 has been found to be upregulated in inflammatory breast tumors [84]. Moreover, it has been suggested that the IL-6/JAK2-STAT3 axis enables chemotherapy resistance in inflammatory breast cancer [7]. Early reports seemed to indicate the existence of functional crosstalk between ER- and IL-6-activated STAT3 [85,86].

However, more recent research has shown that IL-6 activates STAT3 in ER-expressing breast cancer cells and increases its invasive and metastatic activity. The authors found that IL-6-activated STAT3 drives a different transcriptional program by hijacking shared ER enhancers, and thus activating an oncogenic profile independent of ER. Interestingly, standard ER-targeted therapy was unable to inhibit IL-6/STAT3-induced metastasis, whereas incubation with the JAK1 and JAK2 inhibitor ruxolitinib blocked IL-6/STAT3 activation and in vivo cell invasion, strongly suggesting a functional decoupling of IL-6/STAT3 from ER. The authors also proposed that the expression of ER and phosphorylated STAT3 (pSTAT3) may be considered independent prognostic factors in breast cancer, and that targeting IL-6/STAT3 could have clinical potential in ER-positive, endocrine therapy-refractory patients [87].

IL-6 is a pleiotropic cytokine. Indeed, IL-6 is implicated in a number of biological events that control tumor progression. For instance, it has been demonstrated that IL-6, secreted by adipocytes, is able to induce epithelial–mesenchymal transition [88], enrichment of cancer stem cells, and resistance to PI3K inhibitors [89] via STAT3 activation in ER-expressing MCF-7 breast cancer cells. Moreover, the potential of the IL-6/STAT3 axis to promote resistance extends to cyclin-dependent kinase 4/6 (CDK4/6) inhibitors [90] and tamoxifen [91].

A 5-year course of treatment with tamoxifen is the most prevailing endocrine therapy for patients with ER-positive breast tumors. A meta-analysis including 62,923 ER-positive breast cancer patients who were disease-free after 5 years of tamoxifen-based treatment showed that there is risk of recurrence in patients initially responsive to tamoxifen. The cumulative risk of recurrence at 20 years post-treatment ranged from 13% in early-stage patients to 41% in women presenting tumors of 2 to 5 cm with 4 to 9 involved nodes [92]. Thus, it is vital to define the molecules that may mediate and predict responses to tamoxifen. In this regard, Tsoi et al. showed that tamoxifen resistance can be reversed by tocilizumab, an IL-6R-blocking antibody. Moreover, analysis of tumor samples demonstrated that IL-6R expression was significantly associated with tamoxifen resistance and poor overall survival in ER-positive breast cancer patients [93].

In addition, IL-6 has been identified as an essential component of the bone microenvironment. Bone IL-6 can bind to receptors on breast tumor cells, promoting chemotaxis, migration, adhesion, and ultimately the growth of bone metastasis [94,95]. Taken together, this evidence suggests that the IL-6/JAK-STAT3 loop plays a basic role in luminal breast cancer progression and deserves further research.

### 4.2. Activation of the JAK-STAT Pathway by PRL

PRL is a hormone that stimulates breast cell proliferation and milk production in a paracrine fashion by binding to the membrane PRLR [96]. In breast cancer, both PRL and PRLR are extensively expressed [97]. Data from large, prospective studies showed an association between high levels of circulating PRL and invasive breast cancer in postmenopausal women. Interestingly, the association was particularly strong for ER-positive tumors [98,99]. In addition, Sutherland et al. reported that ligand-induced activation of PRLR supports bone metastasis by stimulating osteoclast activity [100].

Signal transduction downstream of PRLR is complex. A number of different receptor-bound kinases have been reported to interact with the cytoplasmic domain of PRLR, including JAK2, SRC, FYN, JAK1, and TEC, activating signals that control cell differentiation, proliferation, survival, and even regulate the cytoskeleton [101]. It is currently known that PRL-PRLR/JAK2-STAT5 is the principal signaling loop mediating the pathological functions of PRL in breast cancer. In fact, an animal model of PRL-induced breast carcinogenesis demonstrated that JAK2 is needed for cancer initiation but not for maintenance of tumors [102]. However, the PRL/STAT5 cascade was proved to interfere with breast cancer-1 (BRCA1)-induced activation of the cyclin-dependent kinase inhibitor, p21, hampering BRCA1 inhibition of the cell cycle and thus allowing cell proliferation as a response to PRL [103]. In line with the observation suggesting that PRL/STAT5 is important for cancer initiation but not for tumor progression [102], Peck et al. showed that levels of nuclear phosphorylated STAT5 are significantly higher in ductal carcinoma in situ than in cases of invasive and metastatic disease. The authors even suggested that the depletion of activated STAT5 is a predictor of poor clinical outcomes and risk of endocrine therapy resistance in patients with breast cancer [104].

On the whole, activation of JAK2-STAT3 and JAK2-STAT5 downstream of IL-6R and PRLR, respectively, may play a coordinated role in the development and progression of luminal breast cancer, suggesting the importance of these signaling cascades in ER-positive breast tumors.

## 5. Adaptor Molecule LNK as an Inhibitor of JAK-STAT

The relevance of the JAK-STAT pathway in breast cancer was further supported by an integrative cancer interactome analysis. This showed that STAT3 is the most central protein governing communication with tumor-linked proteins in the breast cancer network, whereas STAT5a is the only STAT showing a direct interaction with ER [105]. In normal cells, the activity of the JAK-STAT cascade is regulated at different levels. Members of the suppressor of cytokine signaling (SOCS) family of proteins are induced via the JAK-STAT pathway itself and are able to repress the kinase activity of JAK1, JAK2, and TYK2, thus blocking the signaling transduction process [106]. On the other hand, protein inhibitor of activated STAT (PIAS) inhibits STAT transcriptional activity, working either via direct interaction that hinders the union of STAT with DNA or recruiting transcriptional cofactors to STAT target genes [107]. Finally, the JAK-STAT signaling may also be regulated by members of the protein tyrosine phosphatases (PTPs), which maintain the delicate balance between tyrosine phosphorylation and dephosphorylation in cells [108].

In addition, other molecules have been found to regulate the JAK-STAT cascade. One of these negative regulators is LNK. The LNK protein, encoded by the *SH2B3* gene, was first identified in 1995 as a regulator of T-cell activity [109]. Later, structural characterization revealed that LNK was a member of the SH2B family of adaptor proteins and contained three functional domains [110]. The first is a C-terminal SH2 domain of 100 amino acids, which recognizes and binds to target proteins. The second is the N-terminal region of 60 residues, which has the potential to form dimers with other SH2B proteins. The third is a central pleckstrin homology domain (PH) of 120 residues. This allows LNK to localize at the inner side of the cell membrane by interacting with phosphatidylinositol lipids [111]. In human cells, LNK regulates signal transduction downstream of cytokines, growth factors, and hormones by binding specific phosphorylated tyrosine residues via its SH2 domain.

LNK activity is essential for the regulation of hematopoiesis. *SH2B3* transcription is induced by STAT3 and STAT5. LNK is expressed in hematopoietic progenitor cells [112], modulating lymphopoiesis [113,114] and megakaryocytopoiesis [115]. LNK primarily regulates hematopoiesis by binding to phosphorylated JAK2 and JAK3 kinases via its SH2 domain. As a result, it inhibits the activation of downstream transcription factors. However, it has been reported that LNK is also able to bind directly to cytokine receptors, such as EPO receptor, and RTKs like c-KIT and FMS-like tyrosine kinase (FLT3), the latter of which play an essential role in the development of hematological progenitor cells [116]. LNK is subjected to regulation via its interaction with 14-3-3 proteins. These proteins bind to LNK phosphoserine residues 13 (pS13) and 129 (pS129), hindering the attachment of LNK to JAK2 and therefore liberating the signaling cascade [117].

Although the activity of LNK has been mostly described in hematopoietic cells, there is evidence demonstrating the expression and functions of LNK in non-hematological cells, broadening the potential implications of LNK in healthy and abnormal conditions. For instance, LNK is expressed in endothelial cells. Endothelial cell adherence and migration depend upon the formation of focal adhesions (FAs). Interestingly, LNK localizes at FAs, and the inhibition of LNK expression via RNA interference significantly reduces the spread of endothelial cells. In this cell type, LNK is able to regulate β1 integrin-associated pathways, increasing the number of FAs and cell matrix adhesions [118]. In addition, the capacity of LNK to regulate progenitor endothelial cells was demonstrated in vivo. Using *lnk*^−/−^ mice, the authors demonstrated that LNK is a master regulator of cell growth, endothelial commitment, migration, and recruitment for the vascular regeneration of progenitor endothelial cells by inhibiting the activity of cKIT, a tyrosine kinase receptor that activates the JAK-STAT cascade, among others [119].

LNK has also been detected in primary cortical neurons. Notably, it was observed that LNK is able to directly bind to phosphorylated nerve growth factor receptor (NGFR), blocking activation of MEK-ERK 1/2 and PI3K/AKT signaling. As a result, LNK is able to inhibit cortical neuron differentiation and reduce neurite outgrowth [120]. Furthermore, LNK is expressed in neural stem and progenitor cells. Additionally, it can be observed in the subventricular zone of the human brain. After an ischemic stroke, activation of STAT1 and STAT3 induces overexpression of LNK. This mediates a significant reduction in neuronal stem and progenitor cell proliferation in the damaged area [121]. Although the authors did not evaluate the capacity of LNK to inhibit the JAK-STAT cascade in these cells, they indeed suggested that *SH2B3* transcription is regulated by STAT molecules [121].

Finally, there is evidence showing that the expression of LNK is significantly higher in heart samples of dilated cardiomyopathy patients than in normal hearts, and that LNK mediates cardiomyocyte hypertrophy during cardiac remodeling [122]. Again, the authors did not investigate the potential implications of the JAK-STAT pathway in these patients. However, they did find that LNK regulates the FAK-PI3K-AKT-mTOR/GSK3 β cascade and postulated that LNK may also participate in integrin-mediated signal transduction to control cardiomyocyte hypertrophy [122].

Taken together, this experimental evidence suggests that LNK may play a role as a regulator of several cell functions by controlling not only the canonical JAK-STAT cascade but also alternative signaling pathways in a variety of non-hematological cell types, and that further research is needed to fully understand the functions of LNK.

## 6. LNK in Hormone Receptor-Positive Breast Cancer

Since LNK is a major regulator of normal hematopoiesis, it is not surprising that the loss of LNK control is associated with several hematological disorders, myeloproliferative neoplasms [123], and various types of leukemias [124,125,126]. However, the potential role of LNK in solid tumors remains controversial.

Thus far, investigation of LNK has been limited to a small number of solid tumor types. By studying human melanoma tissue arrays and mRNA expression data from available databases, Ding et al. found that LNK is overexpressed in patients with melanoma, and that higher expression of LNK is associated with shorter overall survival [127]. Likewise, elevated expression of LNK was detected in patients with thyroid carcinoma [128]. In glioblastoma patients, elevated expression of LNK predicted worse survival [129]. Similarly, in silico analysis and immunohistochemical study of tissue arrays demonstrated increased LNK expression in patients with high-grade ovarian cancer [130]. Unlike the abovementioned tumors, studies of colorectal and lung cancer showed a negative correlation with LNK expression. In a small study, including tumor samples from 32 colorectal carcinoma patients, it was observed that the expression of LNK was significantly lower than that in adjacent normal tissue [131]. In the case of lung cancer patients, low expression of LNK was found to be correlated with poor prognosis [132].

Overall, analyses of LNK expression in solid tumor biopsies suggest that the level of expression of LNK might be tissue-specific. Interestingly, functional analysis in cell lines has also shown that LNK-associated activities seem to depend upon cell type. For instance, the induction of LNK overexpression in thyroid carcinoma-derived [128] and glioblastoma-derived [129] cell lines resulted in increased cell proliferation and migration, as well as protection against apoptosis. These observations agree with clinical data showing an association between LNK overexpression and poor survival in glioblastoma patients [129]. Likewise, overexpression of LNK in ovarian cancer-derived cell lines restrained apoptosis, while inhibiting LNK expression significantly reduced cell proliferation. These observations support the idea of a potential oncogenic role of LNK in ovarian cancer and might be associated with the fact that patients with advanced ovarian cancer show elevated levels of LNK expression [130]. In sharp contrast, a low level of LNK expression has been found in colorectal and lung tumors. Notably, inducing the overexpression of LNK in colorectal-derived [131] and lung cancer-derived [132] cell lines reduced the proliferation and invasive potential of both cell types. In the case of lung cancer, it was demonstrated that LNK modulated cell activities by suppressing the JAK2-STAT3 and SHP2/Grb2/PI3K/AKT signaling loops [132]. Taken together, these data seem to suggest that the role of LNK during cancer development is tissue type-specific. Nevertheless, data remain scarce. Thus, more research is needed to probe this idea.

To date, very little has been reported regarding the expression and potential function of LNK in breast cancer. Current evidence supporting the potential activity of LNK in luminal breast cancer is outlined in Figure 1. The first report establishing an association between LNK and breast cancer was published in 2015. The authors conducted a cross-cancer analysis of genomic variants in the inflammation pathway using public gene expression datasets. They found that the *SH2B3* missense variant (rs3184504) was associated with breast cancer [133]. In line with former reports, a large in silico analysis of genomic data contained in the UK Biobank showed that the *SH2B3* missense variant was associated with breast cancer [134]. Unfortunately, none of the studies stratified cases by subtype, and they did not establish correlations with clinical outcome. Since the abovementioned missense single-nucleotide polymorphism results in the substitution of tryptophan (Trp) for arginine (Arg) in the PH domain, it has been postulated that it might modify the capacity of LNK to localize at the inner side of the cell membrane, thus hampering interaction with JAK2. Failing to control JAK2 signaling may produce an overactivation of cytokine-induced cascades. In fact, Alexander et al. used CRISPR-Cas9 to create mice that were homozygous for either the Trp or Arg allele. They observed that Trp/Trp LNK was significantly less repressive of IL-12-induced STAT4 phosphorylation compared to Arg/Arg LNK [135]. As previously mentioned, IL-6 and PRL are important inducers of JAK2 activation in ER-positive breast tumors. Thus, evaluating the prevalence of LNK variants in ER-positive tumors would improve our understanding of the participation of LNK in the control of IL-6 and PRL pro-tumoral activities.

Later, an interesting study originally focused on examining LNK expression in triple-negative breast cancer showed that the expression of LNK is significantly higher in triple-negative breast cancer biopsies than in non-triple-negative tumors. Further analysis of breast cancer-derived cell lines demonstrated that the expression of LNK was indeed lower in cells expressing hormone receptors, suggesting that LNK might play different roles in breast cancer subtypes [136]. The impact of increasing the level of LNK expression in tumor cells has been evaluated in vitro in colorectal [131] and lung cancer [132], which has been associated with low expression in corresponding patients. However, up until now, the effect of overexpressing LNK in luminal cancer-derived cell lines has not been explored.

It is important to consider that regulation of the JAK-STAT-LNK axis does not only rely on the level of LNK. Mutations on JAK2, STAT3, and LNK, along with expression of the LNK inhibitor 14-3-3, have also been reported to induce alterations of the signaling cascade. JAK2 gain-of-function mutations, leading to constitutive phosphorylation of STAT3, STAT5, and JAK2 itself, have been recognized in a high proportion of myeloproliferative neoplasms [123]. Nevertheless, the scenario in breast cancer appears to be more complicated. On the one hand, JAK2 amplifications have been mainly identified in triple-negative tumors [137]. In contrast, whole-genome sequencing analysis of 170 breast cancer patients demonstrated that distant metastasis from ER-positive patients causes JAK2 and STAT3 mutations, producing late inactivation of the formerly active cascade [138]. This suggests that LNK regulation might only be relevant at early stages of luminal cancer development.

LNK adaptor activity is negatively regulated by 14-3-3 proteins. It is generally accepted that expression of the majority of 14-3-3 isoforms is elevated in cancer. However, an evaluation of the expression of 14-3-3 isoforms in breast cancer revealed that expression of the 14-3-3 β isoform was significantly increased in luminal tumors. Additionally, it was significantly associated with poor overall survival and shorter relapse-free survival of patients [139]. Interestingly, is has been reported that 14-3-3 β directly binds to ERα, activating the transcriptional activity of the receptor [140]. This suggests that, in ER-positive breast cancer, a complex interaction between 14-3-3 β, ERα, and LNK might contribute to preserving the oncogenic phenotype by activating ERα and repressing LNK. This potential interaction deserves to be investigated more deeply.

Due to the scarcity of published research in the field of LNK and luminal cancer, we conducted an analysis of the available datasets containing genetic data with the intention of shedding some light on the potential associations among LNK, JAK2, STAT3, STAT5, 14-3-3, and clinical outcomes. Considering that several LNK mutations are associated with myeloproliferative neoplasms [123], we first scrutinized publicly available data in the cBioPortal for Cancer Genomics [141,142], searching for mutations in breast cancer samples. An analysis of sequencing data, including 13 independent studies and nearly 4000 patients (Appendix A), showed that only 18 out of 3607 patients (0.5%) had one or more mutations in the *SH2B3* gene (Figure 2a), suggesting that LNK mutations are not associated with breast cancer. Most LNK mutations have been found in myeloproliferative disorders. Although experimental evidence supports the participation of LNK as a regulator of signaling cascades in cancer cells, there are no reports of LNK mutations with biological significance in solid tumors. Our analysis showed that the frequency of mutations in luminal tumors is very low. However, as we previously proposed, deregulation of LNK function might be produced by a number of factors involving other components of the JAK-STAT-LKN axis.

We then investigated the level of expression of LNK-encoding gene *SH2B3* by interrogating the TCGA PanCancer transcriptomic dataset through the cBioPortal platform. Samples were stratified by cancer subtype and specific mRNA expression in 499 luminal A and 197 luminal B samples was compared with expression in normal adjacent tissues. Gene overexpression was considered for samples with a z-score relative to normal adjacent tissue > 2, whereas underexpression was considered for samples with a z-score relative to normal adjacent tissue < 2 (Figure 2b). In order to establish potential associations with the clinical outcome of luminal cancer patients, we downloaded standardized survival data from cBioPortal to determine whether over- and underexpression of *SH2B3* are associated with overall survival of hormone receptor-positive breast cancer patients. The results presented in Figure 2c indicate that luminal A patients with overexpression of *SH2B3* showed a trend of having reduced 5-year overall survival, whereas patients with *SH2B3* underexpression showed a trend of having increased 5-year overall survival. However, the values did not reach statistical significance. These data seem to indicate that, although the overexpression of *SH2B3* was only observed in a small number of luminal A patients, it seems to predispose them to diminished survival, whereas patients with *SH2B3* underexpression appear to have better prognosis. However, the relatively small number of patients in each group prevented us from reaching a categorical conclusion. Unlike those with luminal A breast cancer, patients with luminal B breast cancer did not show any difference between unaltered and altered expression levels of *SH2B3* (Figure 2d). These observations suggest that the expression of LNK may be downregulated in a proportion of luminal A breast cancer cases, and that reduction of protein levels may be associated with poor prognosis.

LNK activity is regulated by 14-3-3 proteins [117]. There are seven 14-3-3 isoforms in humans. Thus, we first investigated the expression of all seven isoforms in a cohort of luminal A and luminal B samples from the TCGA transcriptomic dataset. As seen in Figure 3a, 14-3-3 isoforms ζ and β were overexpressed in 30% and 20% patients, respectively. Thus, to establish potential associations of 14-3-3 ζ and β isoform expression with the clinical outcomes of luminal cancer patients, we downloaded standardized survival data from cBioPortal. Patients showing elevated 14-3-3 ζ expression had significantly reduced 5-year overall survival compared to patients with unaltered expression (Figure 3b). This observation agrees with previous reports showing that 14-3-3 ζ is overexpressed in breast cancer and is associated with a poor prognosis [143]. Data showed that the second most frequently overexpressed isoform was 14-3-3 β. Of note, survival analysis demonstrated that patients presenting 14-3-3 β overexpression had a significantly lower probability of 5-year overall survival than patients with unaltered expression. The difference was even more significant than that observed for the 14-3-3 ζ isoform. This is relevant because 14-3-3 β is the only isoform that has been directly associated with luminal breast cancer since it interacts with and activates the transcriptional potential of ERα [140]. These findings suggest that overexpression of LNK regulators plays a detrimental role in luminal breast cancer. The 14-3-3 β isoform is of particular interest because it may play a dual role in luminal cancer by activating the oncogenic action of ERα and by blocking the inhibitory activity of LNK, favoring cell proliferation and resistance to treatment [140,143]. It is necessary to probe this hypothesis via experimental work using ERα-expressing cell lines.

## 7. Conclusions

Hormone receptor-positive breast cancer is still a serious global health problem [1]. Treatments have been designed to inhibit the activity of ER or the production of estrogens. However, a proportion of patients develop resistance during treatment. Thus, exploring different signaling cascades that play roles during breast cancer development may provide new therapeutic targets. The JAK2-STAT3 and JAK2-STAT5 pathways have been demonstrated to be constitutively active in luminal breast cancer via the stimulation of IL-6 [87,88,89,90,91,92,93,94] and PRL [101,102,103], respectively. The activity of the JAK-STAT signaling cascade promotes proliferation, migration, invasion, and the acquisition of resistance. In some cell types, the JAK-STAT pathway is inhibited by the adaptor protein LNK. However, in hormone receptor-positive breast cancer, the potential activity of LNK seems to be part of an intricate landscape of interactions that deserves to be further explored. In this sense, underexpression of the LNK-encoding gene *SH2B3* is more frequently observed in luminal A cases than in luminal B cases. As the genetic profile of luminal A and luminal B tumors is different, it will be important to perform functional studies using cell lines derived from luminal A tumors to demonstrate whether downregulation of LNK expression is related to luminal A-associated proteins, and if this has an impact on the activity of JAK-STAT transduction signaling.

On the other hand, LNK activity is regulated by 14-3-3 proteins. Isoforms of 14-3-3 can bind to and regulate many target proteins, thus playing important roles in the course of pathological disorders. Here, we detected that two 14-3-3 isoforms are overexpressed in hormone receptor-positive breast tumors. In particular, 14-3-3 ζ and 14-3-3 β overexpression was significantly associated with diminished 5-year overall survival. The overexpression of 14-3-3 ζ has been detected in many tumor types, but it is particularly challenging to address 14-3-3 β in the context of luminal breast cancer because it is able to interact and activate ERα. However, it is not presently known whether this isoform can also block LNK activity. Research focused on the mechanisms of 14-3-3 isoforms as negative and positive regulators of relevant signaling cascades in luminal breast cancer may soon provide new therapeutic targets.

Finally, the presence of an *SH2B3* missense variant, with a potentially diminished capacity to interact with the cell membrane, might also contribute to regulating LNK activity in breast cancer. The present review covers a rather novel subject. However, due to the relevance of the JAK-STAT signaling pathway in luminal breast cancer, we consider it important to assemble current information on the involvement of LNK in the control of this cascade in order to start building innovative models of regulation for this type of cancer. It is clear that research is still needed to fully understand the participation of LNK in the development and progression of hormone receptor-positive breast cancer. The information presented herein will also help to visualize future directions for investigation that may provide data to elucidate the possibility of designing novel therapeutic strategies for hormone receptor-positive breast cancer based on the regulatory mechanism of LNK.

## Figures and Tables

**Figure 1 ijms-24-14777-f001:**
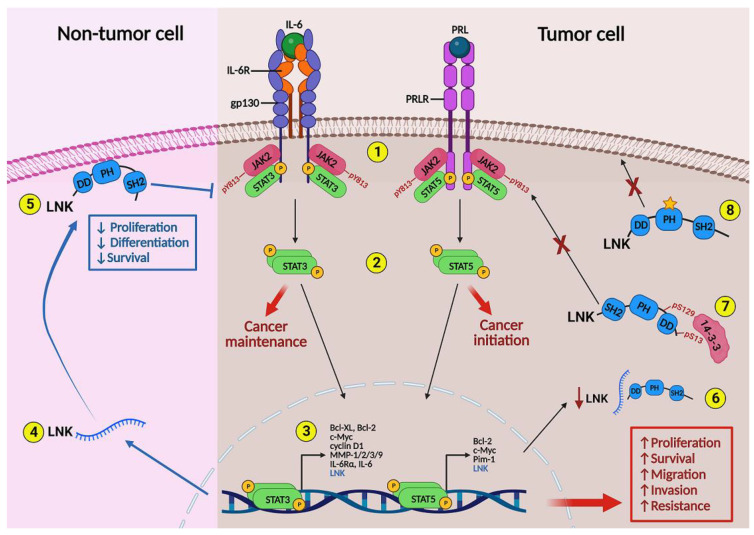
Regulation of the JAK-STAT signaling pathway by LNK in hormone receptor-positive breast cancer and non-tumor cells. Expression of IL-6R/gp130 and PRLR in non-tumor breast cells is closely associated with mammary gland development, whereas it is associated with cancer maintenance and initiation in hormone receptor-positive tumor cells. (**1**) Binding of IL-6 and PRL to their receptors induces activation of receptor-attached JAK2 kinases. (**2**) Active JAK2 phosphorylates STAT3 and STAT5 transcription factors. (**3**) STAT3 and STAT5 homodimers translocate into the nucleus and induce transcription of a number of genes that stimulate proliferation, differentiation, and survival of mammary cells. (**4**) STAT3 and STAT5 also stimulate the expression of LNK. (**5**) LNK localizes to the inner side of the cell membrane via interaction of the PH domain with membrane phosphatidylinositol lipids. This allows binding of the SH2 domain to JAK2, inhibiting the cascade. In tumor cells, constitutive activation of the IL6-IL-6R/JAK2-STAT3 and PRL-PRLR/JAK2-STAT5 loops may be associated with dysfunction of LNK. Three potential mechanisms have been proposed: (**6**) a reduction in LNK expression; (**7**) a blockage of LNK activity by binding with the inhibitor molecule 14-3-3; and (**8**) the presence of the *SH2B3* missense variant rs3184504 (yellow star), which may be unable to interact with the cell membrane. As a result, increased proliferation, survival, migration, invasion capacity, and resistance to endocrine treatment have been reported. IL-6: interleukin-6; IL-6R/gp130: interleukin-6 receptor/glycoprotein 130; PRL: prolactin; PRLR: prolactin receptor; JAK2: Janus kinase 2; STAT3: signal transducer and activator of transcription 3; STAT5 signal transducer and activator of transcription 5; LNK: lymphocyte adaptor protein; PH: pleckstrin homology domain; SH2: Src homology 2 domain; DD: dimerization domain; pS13: phosphoserine 13; pS129: phosphoserine 129. Figure created with BioRender.com.

**Figure 2 ijms-24-14777-f002:**
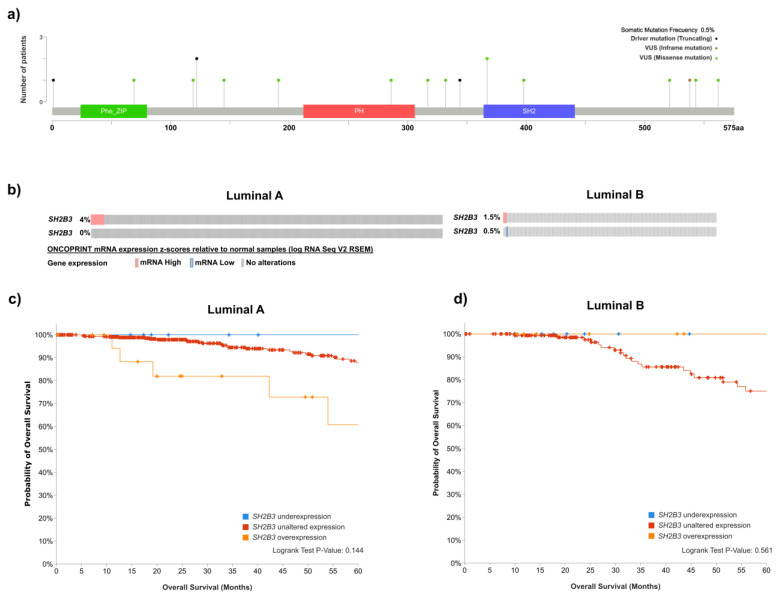
Analysis of expression and mutations of LNK. Publicly available data were downloaded through the cBioPortal for Cancer Genomics. (**a**) Analysis of sequencing data, including nearly 4000 breast cancer patients. Localization of 18 mutations detected. (**b**) Oncoprint of the expression of LNK-encoding gene *SH2B3*, investigated in luminal A and luminal B tumor samples from the TCGA transcriptomic dataset including 499 cases of luminal A and 197 cases of luminal B. Gene overexpression (mRNA High) was considered for samples with z-scores relative to normal adjacent tissue > 2, whereas gene underexpression (mRNA Low) was considered for samples showing z-scores relative to normal adjacent tissue < 2. Association of *SH2B3* under- and overexpression with 5-year overall survival of patients with luminal A (**c**) and luminal B (**d**) breast cancer. Standardized survival data from cBioPortal were downloaded; a statistical comparison of samples showing overexpression (EXP > 2) with samples presenting unaltered expression is presented.

**Figure 3 ijms-24-14777-f003:**
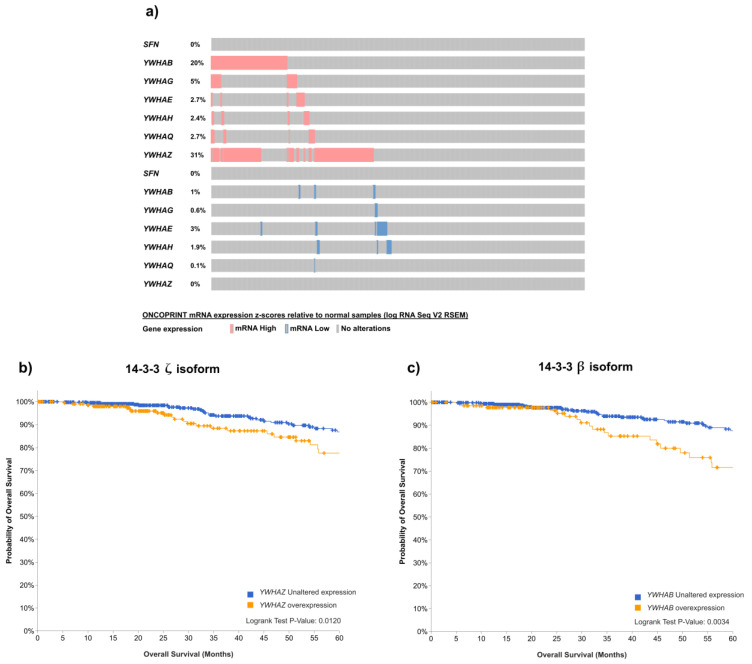
Effect of 14-3-3 isoform expression on clinical outcomes. Publicly available data were downloaded through the cBioPortal for Cancer Genomics. (**a**) Oncoprint of the expression of seven human 14-3-3 isoforms in luminal A and luminal B tumor samples from the TCGA transcriptomic dataset, including 499 cases of luminal A and 197 cases of luminal B breast cancer cases. Individual genes are represented as rows, and individual cases are represented as columns. Gene overexpression (mRNA High) was considered for samples with a z-score relative to that of normal adjacent tissue > 2, whereas gene underexpression (mRNA Low) was considered for samples showing a z-score relative to normal adjacent tissue < 2. Association of 14-3-3 ζ (**b**) and 14-3-3 β (**c**) isoform overexpression with 5-year overall survival. Standardized survival data from cBioPortal were downloaded, and a statistical comparison of samples showing overexpression (EXP > 2) with samples presenting unaltered expression is presented. *SFN*: gene encoding 14-3-3σ; *YWHAB*: gene encoding 14-3-3β; *YWHAG*: gene encoding 14-3-3γ; *YWHAE*: gene encoding 14-3-3ε; *YWHAH*: gene encoding 14-3-3η; *YWHAQ*: gene encoding 14-3-3θ; *YWHAZ*: gene encoding 14-3-3ζ.

**Table 1 ijms-24-14777-t001:** Breast cancer classification.

Molecular Subtyping Classification	Immunohistochemical-Based Analysis	Genetic Modifications	Frequency	Treatment
[22]	[25]	[22]	[23,24,25,26]	[24,25,26]
Luminal-A	ER+, PR ≥ 20%, HER2−, Ki67 < 20%	Alterations in gene expression: *ESR1*, *GATA3*, *FOXA1*, *XBP1.*	~75%	Hormonal therapy
Luminal-B	ER+, PR < 20%, HER2+, Ki67 ≥ 20%	Gene mutations: *PIK3CA*, *ESR1*, *ERBB2*, *ERBB3.*		Hormonal therapy Chemotherapy
HER2-enriched	ER−, PR−, HER2+	Gene amplifications: *ERBB2*, *GRB7*, *TOPO2*, *MYC* Gene mutations: *PIK3CA.*	15–20%	Targeted therapy (anti-HER2 antibodies)
Basal-like	ER−, PR−, HER2−	Gene mutations: *TP53*, *BRCA*, genetic instability.	10–20%	Chemotherapy (specific therapies are not available)

## Data Availability

Not applicable.

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
