# Peer review of "Modulation of JAK-STAT Signaling by LNK: A Forgotten Oncogenic Pathway in Hormone Receptor-Positive Breast Cancer"

_ijms, 2023, doi:10.3390/ijms241914777_

Round 1

Reviewer 1 Report

This review prepared by Lopez-Mejia et al. provides a well-written overview of the health burden of breast cancer, a summary of the different breast cancer subtypes, as well as more specific details on the involvement of JAK-STAT signalling in breast cancer. Specifically, the authors focus on the potential role of the protein LNK in the regulation of the JAK-STAT pathway and the implications in breast cancer, where they mine publically available data to assess factors such as LNK mutations, expression of LNK and JAK-STAT family members and correlations to survival of breast cancer patients. While the authors provide a logical summary of these topics, there are some key aspects that need addressing before the manuscript is suitable for publication.

Major comment:

Given that the manuscript’s main focus is the regulation of JAK-STAT signalling by LNK in breast cancer, the authors do not discuss or provide strong evidence that this signalling regulation is very relevant in breast cancer. LNK has well established roles in regulating JAK-STAT and other signalling pathways in hematopoietic cells, however outside of this system, the authors provide some literature of LNK’s roles in other cell types but not related to directly regulating JAK-STAT signalling (paragraph on lines 428-442). If expression of LNK is low/not relevant in other tissues, such as breast, then it is not clear what role it would play there. The literature regarding the role of LNK in other solid tumors is variable; the authors state what is known but do not attempt to provide a discussion on potential reasons behind this variability (again, tissue expression? Expression of 14-3-3? Relevance of potential LNK targets in the cancer type?). The literature on LNK function in breast cancer is sparse and seemingly limited to correlative studies implicating LNK mutations or expression levels. Again, the authors state that LNK expression is different in different breast cancer subtypes but do not go further in depth to postulate what this could mean (tumor suppressor vs oncogene in different settings? Involvement of hormones and signalling pathways? Linking to different roles of JAK-STAT in the different breast cancer subtypes?). As reference #132 seems to be the only main study linking these pathways in breast cancer, the authors should discuss the findings of this paper in much greater detail in the text (not only summarizing in figure 3, which can be mistaken as simply a hypothesis from the authors if these details are not also discussed in the text). Finally the authors mine public breast cancer data and conclude that LNK mutations are not frequent, LNK is most often not altered in expression (and more often overexpressed than underexpressed), and that LNK overexpression may be correlated with worse survival but patient numbers are too low to conclude (the analyses here should be re-done, see comment below). Overall, this leaves the reader confused as the literature states that LNK should be lower expressed in hormone-receptor positive breast cancer types. It is also unclear if this overexpression of LNK, which may correlate with survival, is linked to the JAK-STAT pathway at all (literature on other solid tumors where LNK is overexpressed do not refer to any involvement of JAK-STAT signalling in this role - lines 449-462). Indeed, LNK can also regulate other signalling pathways. Overall, the authors need to address these discussion points and make much stronger conclusions at the end to clearly indicate now what is the current knowledge and whether the regulation of JAK-STAT signalling by LNK is really relevant in breast cancer or whether LNK may have other oncogenic roles (or not). Based on the current form of the manuscript and the details given, this reviewer does not feel that the topic indicated by the title is properly depicted or supported by the literature, but perhaps if the authors address these points and add additional literature details and discussion, this may change.

Minor comments:

11.  It is not clear to this reviewer how the survival analyses were performed: expression of all genes are shown on one graph but one “unaltered group” is shown together with one statistical analysis. Does this mean the patients in this unaltered group have no overexpression of any of the genes? It is not clear how it can be concluded which genes correlate with survival this way, apart from by eye. Some are above or below the unaltered group, and yet one significant p-value is given. It would seem much more logical to perform the analyses with the genes separately and be able to perform individual statistical analyses. Furthermore, the number of the patients (and hence the power of the analysis) would be increased if they were split into “low” and “high” expression for each gene, based e.g. on median expression or upper/lower quartiles. The authors could also look for expression of 14-3-3 isoforms in an attempt to indicate whether this could be a reason for LNK function being blocked. The authors should conduct these new analyses, and based on the results, they should indicate much more clearly what they conclude from these data regarding the role of LNK (or individual JAK-STAT members) in breast cancer, which as discussed above is currently far too brief (lines 558-561).

2.       The data in Figure 4c are not convincing. Firstly, the main activating phospho-tyrosine sites for STAT5A/B (Y694/Y699) are not included, which would be the main indication of activity in the cancer cells. Secondly, much of the heatmap is blue, does this mean the STATs are not phosphorylated? The role of JAK-STAT signalling is established in breast cancer, the authors should keep their focus and analyses mainly to assessing the role of LNK.

3.       The prevalence of infiltrating duct carcinomas and invasive lobular carcinomas are indicated (lines 138-139), but the authors should state what subtypes make up the remaining 10-20% of breast cancers for completeness.

4.       On lines 207-210 it is stated that “nuclear ERbeta2 correlated with disease-free survival, and predicted response to hormonal therapy”. The authors should indicate more specifically whether the nuclear protein levels were increased/decreased and whether survival or therapy response was increased/decreased.

5.       Line 244 - inactive STATs are bound as anti-parallel dimers and upon activation, they undergo a conformational change to form parallel dimers. This should be accurately indicated. Similarly, the authors should correct that the STAT dimers (at least the active parallel dimers) form from interactions between the SH2 domain and the activating phospho-tyrosine (not the N-domain, correct lines 272-274).

6.       Line 271 - the authors missed STAT4.

7.       Lines 372-374 - The beginning of this sentence should be re-worded as it implies that PRL-STAT5 promotes cancer and yet the study demonstrates that it acts as a tumor suppressor.

8.       Lines 384-385 - the authors state that “one of the most important negative regulators of the JAK-STAT pathway is LNK”, this is not true and heavily biased. The authors should also briefly discuss e.g. SOCS proteins, protein tyrosine phosphatases and PIAS as negative regulators of the JAK-STAT pathway, before continuing to focus on LNK.

9.       Line 428 - remove the word “target”.

10.   Line 558 - should be Figure 5 instead of 4.

11.   Line 586: “in normal cells” should be changed to “in some cell types” - it is not clear that LNK regulates JAK-STAT in all normal cell types.

12.   Throughout the manuscript, gene names should be written in italics (including reference to mRNA, also in figures).

The manuscript is overall well written but has a number of minor spelling and grammar mistakes throughout, which should be looked over and corrected by a native English speaker.

Reviewer 2 Report

It is a well-written review paper on the role of JAK-STAT signaling in hormone receptor-positive breast cancer. I have a few minor suggestions for improvement:

1. The introduction appears to be too lengthy. The background information on breast cancer, including sections 1 and 2, spans two pages. Additionally, Figure 1 does not seem essential and can be adequately described in the text. Therefore, I recommend deleting Figure 1.

2. It would be helpful to include a table summarizing the subtypes of breast cancer for section 3.

3. Why there are two complexes of IL-6/gp130 and PRL/PRLR in Figure 2 and that makes it difficult to highlight the feedback. In addition, Figures 2 and 3 appear redundant and could be removed.

4. The section heading for "Conclusion" should be numbered as 9.

5. When referring to the contributions of the authors, it would be appropriate to reference the relevant "sections", indeed of "chapters".

Reviewer 3 Report

No obvious recent reviews on the broader JAK-STAT pathway in breast ca found. DOI: 10.1016/j.lfs.2022.120996 – focuses on STAT3 alone.

DOI: 10.2174/0929867328666201207202012 – concentrates more on therapeutic targeting

The authors present a relatively detailed review of all JAK and STAT molecules in breast cancer as well as their interaction with LNK including some new data analyzing on-line datasets with respect to these molecues.

Abstract

The abstract is well written and a sloid summary.

However, the first 4-5 lines giving a basic introduction to ER positive breast cancer could be reduced to 1-2 lines to allow room for a modestly more detailed summary of review contents.

1.       Introduction

Introduction is well written and provides a good rationale for the review. A few comments:

31 - Breast and cervical tumors are two of the most important types of female cancer – important is an unusual choice of word. Cervical cancer cases are falling with vaccinations in young women.

34 - Note early detection of cervical cancer does not reduce cases, but does affect mortality.

48 – ‘regulate not only the function of ER+/PR+ cells, but also 48 ER-/PR- cells’ suggest that this statement should be referenced.

54 – 60 – this paragraph is a little vague on resistance and need for therapies. Suggest it should be stated in terms of both early cancer (ie relapse on or after adjuvant anti-estrogens) and metastatic cancer (progression on current anti-estrogen-based treatment). Probably should also mention in passing that duel blockade of ER alongside CDK 4/6, mTOR and PI3Kinase has also yielded benefits but still with eventual resistance in most patients.

2.       BrCa worldwide

I do not understand the function of this section in this review – breast cancer incidence and sociocultural and geographic variations appear entirely unrelated. Inevitably more new agents being developed for breast cancer is only likely to initially widen gaps in care and outcome due to rising costs.

3.       Subtypes.

Again, although this is well-written it is material widely covered in many focused publications and so appears to bulk out this review without significant contribution. A brief introduction of the concept of sub-types in the intro to preface the review should suffice.

4.       ER/PR-Associated Carcinogenic Cell Signaling Cascades

This section is well written but the the discussion of classical and non-classical ER pathways does not interface with the later review material on JAK/STAT and Lnk.

Again the ER beta and GPER1 paragraphs do not have clear relevance to the review topic as they do not tie in to JAK-STAT signaling in breast ca later in the review.

194 – 195 note CCND1 is the gene that produces cyclin D1 so both don’t need listing.

202 – it is probably something of an overstatement to imply that ER alpha is fully understood.

208 – ‘nuclear ERβ2 and ERβ5 208 correlated with overall survival’. Should say whether the correlation was with better or worse OS.

220 ‘non-hormone responsive breast tumors’, do you mean hormon resistant here?

5.       JAK-STAT pathway

This is probably where the main review should start – sections 2 – 4 do not materially contribute.

240 – ‘The JAK-STAT signal is activated after binding of ligands to classical receptors and RTKs.’ This sentence and the following couple of lines are a bit vague as a basic intro to the core topic of the review. Suggest explain a bit more specifically.

260 – ‘JAK2 is activated in hormone and growth factor receptors’. Not sure what is meant here.

270 - ‘The family of the STAT transcription factors is composed of seven different proteins: 270 STAT1, STAT2, STAT3, STAT5A, STAT5B, and STAT6.’ This list is six proteins, STAT4 has been omitted.

272Suggest mentioning briefly whether STAT proteins have established consensus DNA binding sequences in promotors or enhancers of their target genes.

The JAK and STAT family paragraphs are otherwise good.

6.       JAK-STAT in ER pos Breast ca

334 – ‘The authors also proposed that expression of ER and phosphorylated STAT3 (pSTAT3) may be considered independent prognostic factors in breast cancer’. Suggest expanding on this to explain more the content of this reference.

342 – ‘Taking into account that tamoxifen is the most prevailing treatment for patients with ER-positive breast tumors, and that about 50% of them will suffer recurrence, it is vital to define the molecules that may mediate and predict response to tamoxifen.’ I am uncertain of the origin of the 50% figure but cannot think of a scenario where it is accurate. Are you referring to early breast cancer where from the 15 year Oxford overview around 30% relapsed by 15 years (DOI: 10.1016/S0140-6736(05)66544-0) or to metastatic disease where 90%+ eventually relpase?

366 – ‘mediating the physiological functions of PRL in breast cancer’. Do you mean pathological or pathophysiological functions?

372 – ‘In line with the participation of PRL-STAT5 in the development of cancer, Peck et al. showed that depletion of activated STAT5 is a predictor of poor clinical outcome.’ If PRL-STAT5 promotes cancer wouldn’t you expect depletion to be a predictor of good outcome?

Figure 2 is pretty good but suggest lightening the colour of some molecules to make molecule names easier to read and slightly enlarge the fonts.

468 – ‘, the expression of LNK was found to be low, and it correlated with poor prognosis in lung cancer patients.’ Did high or low LNK correlate with poor prognosis here?

Prose generally well written. Occasional grammatical errors.
